# Computational Study for the Unbinding Routes of β-*N*-Acetyl-d-Hexosaminidase Inhibitor: Insight from Steered Molecular Dynamics Simulations

**DOI:** 10.3390/ijms20061516

**Published:** 2019-03-26

**Authors:** Song Hu, Xiao Zhao, Li Zhang

**Affiliations:** Department of Applied Chemistry, College of Science, China Agricultural University, Beijing 100193, China; husong@cau.edu.cn (S.H.); zhaoxiao@cau.edu.cn (X.Z.)

**Keywords:** O*f*Hex1, unbinding mechanism, “open-close” mechanism, steered molecular dynamics, PMFs

## Abstract

β-*N*-Acetyl-d-hexosaminidase from *Ostrinia furnacalis* (O*f*Hex1) is a new target for the design of insecticides. Although some of its inhibitors have been found, there is still no commercial drug available at present. The residence time of the ligand may be important for its pharmacodynamic effect. However, the unbinding routes of ligands from O*f*Hex1 still remain largely unexplored. In the present study, we first simulated the six dissociation routes of *N,N,N*-trimethyl-d-glucosamine-chitotriomycin (TMG-chitotriomycin, a highly selective inhibitor of O*f*Hex1) from the active pocket of O*f*Hex1 by steered molecular dynamics simulations. By comparing the potential of mean forces (PMFs) of six routes, Route 1 was considered as the most possible route with the lowest energy barrier. Furthermore, the structures of six different states for Route 1 were snapshotted, and the key amino acid residues affecting the dissociated time were analyzed in the unbinding pathway. Moreover, we also analyzed the “open–close” mechanism of Glu368 and Trp448 and found that their conformational changes directly affected the dissociation of TMG-chitotriomycin. Our findings would be helpful to understanding and identifying novel inhibitors against O*f*Hex1 from virtual screening or lead-optimization.

## 1. Introduction

As an important structural constituent of insects, chitin is a high polymer of β-1,4-linked *N*-acetyl-d-glucosamine (GlcNAc) [1]. Insects experience several processes of molting during their growth and development, including the synthesis and degradation of a large amount of chitin [2]. Two members of the glycoside hydrolase family are required during chitin degradation. One is GH20 β-*N*-acetyl-d-hexosaminidase (Enzyme Commission (EC) 3.2.1.52), which participates in the hydrolysis of the glycosidic bonds of glycoproteins, glycans and glycolipids [3,4]. Because chitin is absent in humans, the enzymes related to chitin hydrolysis are potentially safe for humans, which can be used as specific targets for the exploitation of efficient and green pesticides [5].

The structures of crystal complexes of multiple GH20 β-*N*-acetyl-d-hexosaminidase enzymes have been resolved in recent decades. Among them, SmChb was the first identified structure from *S. marcescens* [6], followed by HsHexA and HsHexB from humans [7] as well as O*f*Hex1 from *Ostrinia furnacalis* [8]. O*f*Hex1 has been intensively studied in recent years due to the development of new insecticides. A large number of its inhibitors have been found, such as *N*-acetyl-d-galactosamine-thiazoline (NAG-thiazoline) [9], (O-(2-acetamido-2-deoxy-d-glucopyranosylidene)amino-*N*-phenylcarbamate) (PUGNAc) [10], Q2 [11] and TMG-chitotriomycin [12], and their crystal structures in complex with O*f*Hex1 have been all clarified [8,11,13,14]. Among them, TMG-chitotriomycin (Figure 1a) is a mimetic compound of substrate (GlcNAc)_4_, in which *N*,*N*,*N*-trimethyl-d-glucosaminyl displaces the end of GlcNAc, leading to highly inhibitory activity against O*f*Hex1 [8]. O*f*Hex1 has a narrow and deep substrate-binding pocket with the −1 and +1 sites. The TMG ring and +1 GlcNAc Ⅰ of TMG-chitotriomycin can firmly bind with −1 and +1 sites, respectively (Appendix A). There are also two additional loops (L314–335 and L478–496) at the entrance of the O*f*Hex1 pocket, which are important for binding. A comparison of the structure [11] between TMG-chitotriomycin-O*f*Hex1 or Q2-O*f*Hex1 complex and free O*f*Hex1 shows that there is an open and a closed active pocket in both structures, which are controlled by the residues Glu368 and Trp448. Liu et al. [13] have found that the binding of inhibitor (3aR,5R,6S,7R,7aR)-5-(hydroxymethyl)-2-(methylamino)-5,6,7,7a-tetrahydro-3aH-pyrano[3,2-d]thiazole-6,7-diol (NMAGT) to O*f*Hex1 is tighter than that of NAG-thiazoline to Trp448 in the closed-state pocket during molecular dynamics (MD) simulations, while the *K*_i_ of NMAGT is much lower than that of NAG-thiazoline, suggesting that the conformation of Trp448 is important. Although the ligand-binding mechanism of O*f*Hex1 has been studied by several crystal complexes to identify the key residues involved in binding, the actual mechanism underlying the ligand unbinding remains unclear.

Steered molecular dynamics (SMD) is a good method to accelerate ligand unbinding with low computational cost and reproducibility [15]. Moreover, the biological effects of the protein binding of drugs are influenced by the trends of the drug dissociation. SMD has been applied to successfully study the dissociation of some inhibitors, such as VEGFR2 inhibitors, kinase inhibitors and B-RAF inhibitors and insulin [16,17,18,19,20]. Therefore, SMD can be useful in studying model ligand–receptor unbinding pathways. Furthermore, the potential of mean force (PMF) is important for exploring the mechanism underlying the ligand unbinding [21]. Several methods are used to extract energy values, such as umbrella sampling [22], free energy perturbation [23], and thermodynamic integration [24]. However, these methods are expensive and time-consuming. Besides these methods, Jarzynski’s equality is a good approach to extract energy from SMD simulations with good efficiency, accuracy, and ease of use [25].

In the present study, the complex structure of the dimer O*f*Hex1 with TMG-chitotriomycin was selected to explore the unbinding mechanism of TMG-chitotriomycin away from O*f*Hex1 with a combination of conventional MD and SMD. First, three parallel MD simulations were applied to find the initial conformation of SMD. Second, the CAVER software was used to analyze possible unbinding routes. Lastly, we performed SMD on six different directions and constructed the PMFs by Jarzynski’s equality. Based on these analyses, some important residues and dissociation mechanisms were discussed, which would be beneficial for designing a novel O*f*Hex1 inhibitor with a long residence time.

## 2. Results and Discussion

### 2.1. The Conformational Change of MD Simulation

The crystal structure of the free O*f*Hex1 reveals a homodimer enzyme with two monomers in the adjacent symmetrical units. The interface between monomers (Figure 1b, chain A and chain B) is formed mainly by two rigid loops: residues 314–335 (L314–335) and residues 478–496 (L478-496). Our previous research [26] has showed that the loops L314-335 and L478-496 are flexible when the monomer structure of O*f*Hex1 is used as the initial structure in the MD simulation. This is paradoxical in the crystal complex structure, where L314–335 and L478–496 are rigid since L314–335 are stabilized by two pairs of disulfide bonds and L478–496 are parallel to their counterparts by numerous interactions. Therefore, it is necessary to use the dimer structure as the initial structure for the MD simulation. The crystal structure of the monomer O*f*Hex1 with TMG-chitotriomycin (Protein Data Bank (PDB) ID: 3NSN) has been reported by Yang’s group [8]; however, the crystal structures of a homodimer with two TMG-chitotriomycins have not published formally in the PDB. In the present study, we used the crystal complex structures of a homodimer with two TMG-chitotriomycins from our cooperation partner, Yang’s group.

Then, we performed three parallel conventional MD simulations (named CMD1, CMD2, and CMD3, respectively) of the O*f*Hex1-TMG-chitotriomycin complex for 125 ns. The simulation results showed that the ligand (TMG-chitotriomycin) kept its original position in CMD2. However, the ligands were translocated far away from L478–496 in CMD1 and CMD3 (Figure 1c). Then, we extracted the instantaneous structure on the 125th ns and calculated the distance between the O4 atom of the ligand and NE1 atom of Trp490 during the three simulations (Appendix A). The result showed that the distance in CMD2 was stable, which was consistent with the unchanged position of the ligand (Figure 1c). However, the distances in CMD1 and CMD3 changed greatly to 9.18 Å and 7.62 Å from 4.71 Å, respectively (Figure 1d), which caused the strong hydrogen bond interaction between the ligand and Trp490 to be missing.

The root means square deviation (RMSD) of TMG-chitotriomycin and O*f*Hex1 backbone atoms was also calculated to analyze the structural stability from CMD1, CMD2 and CMD3. From 60 ns to 110 ns, the fluctuations of the backbone atoms were large in CMD1 and CMD3. On the contrary, the fluctuation was gentle in CMD2 (Appendix A). To explore the shift ability of receptor residues, the root means square fluctuations (RMSFs) of backbone atoms were calculated (Figure 2). The fluctuations of CMD1 and CMD3 were similar, indicating that there were possibly the same conformational changes for O*f*Hex1. The shift of L314–335 in CMD1 and CMD3 was bigger than that in CMD2. Then, we aligned the three average simulation conformations with the crystal OfHex1 structure, respectively (Appendix A). The backbone of L314–335 in CMD1 and CMD3 displayed a large shift, while the shift of L314–335 in CMD2 was small. Because the movement of TMG-chitotriomycin was far away from L478–496 in CMD1 and CMD3, the conformations of L478–496 were kept stable. Considering the larger tendency of TMG-chitotriomycin away from the active pocket of O*f*Hex1 in CMD1 than that in CMD3, CMD1 was chosen for further study in SMD.

### 2.2. Analysis of Unbinding Pathways by CAVER

CAVER 3.0 was used to analyze the last MD conformation on the 125th nanosecond in CMD1. All possible pathways with different average radii and lengths of the tunnel were identified, leading to a set of nearly 21 pathways. Among all pathways, we selected the top six routes with different dissociation pathways to study, named Route 1, Route 2, Route 3, Route 4, Route 5 and Route 6, respectively (Table 1 and Figure 3a). Route 1 had the widest radius (4.23 Å) and shortest length (2.90 Å). Therefore, its priority reached 0.962, and it might be the most possible route. The priority of the other five routes was 0.906, 0.876, 0.806, 0.879, and 0.730, respectively.

In order to further identify the possibility of the selected routes and the bottleneck residues of all tunnels, 100 snapshots from the last 5 ns of CMD1 were analyzed and clustered to identify all possible pathways. The top six routes of the cluster were analyzed (Appendix A) and compared with those in Figure 3a. The dissociation directions of Routes 1–6 were almost the same, which indicated that the selection of six possible routes was reasonable. Thus, six routes (shown in Figure 3a) were used for the following SMD simulation and PMF analyses. The tunnel residues for TMG-chitotriomycin dissolved from O*f*Hex1 were further confirmed, as shown in Figure 3b. The list of the bottleneck residues obtained by the analysis of the MD trajectory using CAVER 3.0 revealed the structural details of the tunnel gating (Table 2). The most frequent bottleneck in Route 1 tunnel was formed mainly by Trp490 (14%), Trp448 (14%), Tyr475 (11%) and Trp448 (11%). The most frequent bottlenecks in the Route 2 tunnel were formed mainly by Asn489 (11%), Trp490 (11%), Glu526 (11%), Val327 (11%), Glus328 (10%) and Val484 (10%). The most frequent bottlenecks in the Routes 3–6 tunnels are shown in detail in Table 2. The study of all important bottlenecks provides invaluable information about their relative importance and is discussed in detail in Section 2.5.

### 2.3. SMD Simulations

SMD simulations were able to pull TMG-chitotriomycin out O*f*Hex1 along the defined routes. Through careful analysis of the TMG-chitotriomycin-O*f*Hex1 complex, six direction vectors were confirmed, as follows. Route 1 was the direction from the Cα atom of Tyr275 to the ligand of the CAD atom (Figure 4, Route 1). Moreover, the direction of Route 1 was consistent with the rotation trend of ligands observed from CMD1 and CMD3 in Section 2.1. The directions of other routes are described in detail in Section 3.4 and shown in Figure 4.

There were a large number of failed attempts to determine velocity during the SMD simulations, due to the simulation time being too long when setting the velocity to 0.005 Å/ps or the structure being destroyed when setting the velocity to 0.03 Å/ps. Finally, when the velocity was set to 0.015 Å/ps and the force constant was set to 5 kcal·mol^−1^·Å^−2^, the SMD simulations could proceed with a short simulation time and reasonable structure. SMD trajectories were repeated 40 times for all the six routes to reduce statistical error, and then the abnormal data were removed. Finally, 21, 21, 28, 15, 25 and 14 repeat times of the SMD simulations were carried out with the same stiff spring constant for the six directions, respectively. Appendix A shows the curve of the average work profiles versus the distance. The average work profiles were 88.02 ± 9.40, 102.83 ± 11.37, 107.95 ± 11.49, 129.38 ± 12.59, 102.58 ± 11.42 and 121.18 ± 12.18 kcal/mol for Routes 1, 2, 3, 4, 5 and 6, respectively. The data indicated that TMG-chitotriomycin could escape easily from the active pocket of O*f*Hex1 in Route 1, compared with the other five routes. Therefore, we carefully analyzed the changes of the average force versus the distance for Route 1 (Figure 5). In the first 5 Å, the slope of force was the largest, which was dramatically increased to 7 pN, indicating that there was a large resistance to be overcome in the initial stage of unbinding. Then, from 5 Å to 14 Å, the slope of force was reduced, which was dropped to about 4 pN, suggesting that some strong forces were destroyed. After 14 Å, the force became more gradual, which was dropped to around 2 pN. The interactions between TMG-chitotriomycin and O*f*Hex1 along the unbinding route were carefully analyzed as described in Section 2.5.

### 2.4. PMFs during the Unbinding Route of TMG-Chitotriomycin

PMFs were calculated by repeated SMD simulations (21 for Route 1, 21 for Route 2, 28 for Route 3, 15 for Route 4, 25 for Route 5 and 14 for Route 6) through the second-order cumulant expansion of Jarzynski’s equality (Equation (2)), and Figure 6 shows the changes of PMF versus the distance. Route 1 had the lowest energy barrier (approximately 23.66 kcal/mol). The ligand unbindings along Route 2, Route 3, Route 4, Route 5, and Route 6 were all energetically unfavorable, while the energy barriers of them were higher than 30 kcal/mol. When the value of PMF was higher, the dissociation was more difficult. Therefore, Route 1 was supposed to be the most possible unbinding route.

### 2.5. Important Residues during the Unbinding of the Ligand

Figure 7 shows that there are two energy peaks (A and C) related to two high steady-states, two low valley energy wells (B and E) related to the metastable states and a stable state (P) along the unbinding route (Route 1). The state P with the lowest energy is the binding state of TMG-chitotriomycin at the binding pocket. The snapshot of F shows the unbinding state structure with TMG-chitotriomycin exposed to the solvent. The structures of other transition states and metastable states in Route 1 were also extracted from the trajectories along the SMD simulation based on the time.

For Route 1, the PMF showed a sharp increase in the first 300 ps, and the TMG-chitotriomycin-O*f*Hex1 reached the high steady state A, where the carbonyl group of residues Asp367 and Glu368 could form a hydrogen bond with the hydroxy of GlcNAc I of TMG-chitotriomycin, and the indole ring of Trp524 could form an H–π interaction with the TMG ring of the ligand. All these interactions contributed to the energy hill of TMG-chitotriomycin at 300 ps.

Then, as the ligand moved outward from the pocket, TMG-chitotriomycin first arrived at metastable state B (Figure 7), in which the H–π interaction between the indole ring of Trp524 and the TMG ring of ligand was broken. The hydrogen bonds with Asp367 and Glu368 were reserved.

The ligand continued to be pulled out from the TMG-chitotriomycin-O*f*Hex1 and reached the second-high steady state C, which was stabilized through the H–π interaction between the TMG ring of the ligand and the indole ring of Trp448.

For state E, the GlcNAc III of the ligand again formed a hydrogen bond with Glu368, which might be attributed to the ligand being too long and having great flexibility. Trp448 also obstructed the further movement of the ligand. Finally, TMG-chitotriomycin completed the dissociation with state F.

### 2.6. The “Open–Close” Conformational Change of Glu368 and Trp448

Along the unbinding route of TMG-chitotriomycin, Glu368 almost always formed a hydrogen bond with the ligand, indicating that the residue was not only the catalytic amino acid, but also important for stabilizing the ligand and preventing the ligand’s dissociation from the active pocket. The side chain of Glu368 was also rotated to the solvent from the active site after the ligand was pulled out.

In the analysis process of the structure, we found an interesting change of residue Trp448 (Figure 8). Previous study has shown that the Trp448 of O*f*Hex1 has a “open–close” specialty between the free enzyme and complex structure [8,11]. In this study, we also showed similar findings. When comparing state P and state A, Trp448 showed an outward deflection from P to A, and the active pocket became larger. When comparing state B and C or state E and state F, the Trp448 all folded outward from state B to state C or from state E to state F. Moreover, states P, B and E were at a low-energy state, while states A, C and F were at a high-energy state, suggesting that the complex from the low-energy state to high-energy state needed to overcome the resistance of the Trp448 and deflect Trp448 to the “open” state. This also indicated that Trp448 was important for stabilizing ligand binding to the active pocket and blocking the dissociation of the ligand.

This was consistent with the conformational change of Trp448 found in the crystal structure of O*f*Hex1 complexed with inhibitor TMG-chitotriomycin or Q2 [11]. Therefore, Route 1 might be reasonable and similar to the actual situation. Although several crystal structures have been resolved, and conventional MD is used in the previous studies, they are all static. Collectively, we dynamically observed the switching process of Glu368 and Trpp448 by cthe alculation of SMD for the first time.

## 3. Materials and Methods

### 3.1. Preparation of Starting Structure

First, the crystal structure of TMG-chitotriomycin complexed with monomer O*f*Hex1 (PDB ID: 3NSN) [8] was used as the starting structure for the MD simulation, which caused L478–496 to be flexible. This was obviously unreasonable because L478–496 should be positioned parallel to its counterpart in another adjacent monomer by numerous interactions to keep rigid. Therefore, the dimer structure (chain A and chain B) of O*f*Hex1 was employed, and only TMG-chitotriomycin was retained on chain B. Then, the minimization of the complex structure was held using AMBER10 parameters for the protein and EHT parameters and AM1-BCC charges for small molecules by Molecular Operating Environment (MOE) [26].

### 3.2. MD Simulations

AMBER 12 [24] was adopted for all MD simulations. The TMG-chitotriomycin-O*f*Hex1 complex was prepared with GAFF (generalized AMBER force fields) [27] and an ff99SB-ILDN force field [28]. The parameters of the ligands were assigned by the AM1-BCC (AM1 with bond charge corrections) charges [29], with the ANTECHAMBER module. All missing hydrogen atoms were added by applying the LEaP module. Next, all solutes were solvated in a rectangular periodic box of pre-equilibrated TIP3P [30] water molecules, with a margin distance of 10 Å. Sodium counter ions were placed on grids, with the largest positive coulombic potentials around the complexes used to maintain the electro-neutrality of all of the systems. Subsequently, the system was gradually heated from 0 K to 310 K for 30 ps in the NVT ensemble and equilibrated for 110 ps at 310 K in the NPT ensemble. The pressure of NPT was 1 atmospheric. The cut-off value was set at 10 Å for non-bonded interactions. Finally, the PMEMD program [31,32] was used to run the total system for 125-ns MD simulation. The simulation of TMG-chitotriomycin-O*f*Hex1 complex was repeated for three times and the last frame of the simulation was used as the starting point for the SMD later.

### 3.3. CAVER

The structure of TMG-chitotriomycin-O*f*Hex1 after 125-ns MD simulations was then analyzed by CAVER 3.0 [33]. Ions and water were removed. The position of the ligand was defined as the binding site. A maximum distance of 3 Å and desired radius of 5 Å were set to start the point optimization. The approximation of the additively weighted Voronoi diagram was set to 12. 0.9 Å of the minimal probe radius, and 3.5 Å of the clustering threshold was set to compute the tunnel. The other parameters were set as default values. Six routes were identified finally, namely Route 1, Route 2, Route 3, Route 4, Route 5 and Route6, respectively (Figure 4). The routes were listed in CAVER based on the width and length of the routes.

### 3.4. SMD Simulations

The Amber 12 package was used to perform the SMD simulations in the NPT ensemble. The pressure was 1 atmosphere, and the temperature was kept constant at 310 K. The collected six routes were selected for simulations with constant velocity. The velocity was set to 0.015 Å/ps. The force constant was set to 5 kcal·mol^−1^·Å^−2^. The time step was set as 2 fs. The time of SMD simulations was 1.75 ns based on the exit route. The cut-off value was set to 10 Å for non-bonded interactions. To construct the PMFs, SMD simulations were repeated 21, 21, 28, 15, 25 and 14 times from Route 1 to Route 6 to reduce statistical mistakes, respectively.

From the result of CAVER, we obtained six directions. The specific atoms were set by their possible steered directions, as mentioned below in detail. The direction between each group of atoms corresponded to a direction of stretching:(1)The direction from the Cα atom of Tyr275 to the CAD atom of ligand (Figure 4, Route 1);(2)The direction from the Cα atom of Lys273 to the C7 atom of ligand (Figure 4, Route 2);(3)The direction from the Cα atom of Val500 to the CAZ atom of ligand (Figure 4, Route 3);(4)The direction from the Cα atom of Agr220 to the C5 atom of ligand (Figure 4, Route 4);(5)The direction from the Cα atom of Asp367 to the C1 atom of ligand (Figure 4, Route 5);(6)The direction from the Cα atom of Trp524 to the C1 atom of ligand (Figure 4, Route 6).

### 3.5. PMFs Calculation

PMF is the potential of mean force, which describes an average over all the conformations of interest. This method states the free energy difference between two states, A and B. Jarzynski expressed an equality that relates free energy difference to the work regardless of the speed of the process (Equation (1)) [34]. This means that, by computing the work between the two states in question, and averaging over the initial state, equilibrium free energies can be extracted from non-equilibrium calculations. Therefore, this equality has the ability to calculate free-energies from non-equilibrium processes. Capelli et al. have proposed a second-order cumulant expansion approach of Jarzynski’s equality (Equation (2)) to refactor the PMF, in which F (the Helmholtz free energy) is associated with W and β = (*k*_B_T)^−1^. In Jarzynski’s equality, PMF is shown as a function of the dissociation reaction coordinate, which was set as the distance between the given atom of ligand and specific atom of receptor along dissociation route, shown in detail in Section 3.4.
(1)$$e^{\left(-\beta \Delta F\right)}=<e^{\left(-\beta W\right)}>$$
(2)$$\Delta F=\langle W\rangle -\frac{\delta_{W}^{2}}{2k_{B}T}$$
where $\delta_{W}{}^{2}=<W^{2}>-{\langle W\rangle }^{2}$ and *k*_B_T = 0.6186 kcal/mol.

## 4. Conclusions

In the present study, the dissociation behavior of inhibitor TMG-chitotriomycin of O*f*Hex1 was studied by CMD and SMD simulations. The results of CMD revealed that TMG-chitotriomycin exhibited a dissociation trend translocating away from L478–496. Six different routes for the unbinding of TMG-chitotriomycin from O*f*Hex1 were explored. The results of SMD and PMF calculations displayed that Route 1 had the lowest energy barriers among the six routes, and it might be the most possible unbinding route. Ligand unbinding affected the conformational changes of some residues, especially for Trp448 in the crystal structures. In the process of dissociation, Glu368 almost always formed hydrogen bonds with the ligand, suggesting that it was an important residue for designing new inhibitors with long residence time. On the other hand, the average rupture force of Route 1 was 11.36 pN, which could be a reference to find other inhibitors by the rupture force [35]. In the future, many other inhibitors of O*f*Hex1 will be evaluated by SMD to improve its accuracy and usability in drug design.

## Figures and Tables

**Figure 1 ijms-20-01516-f001:**
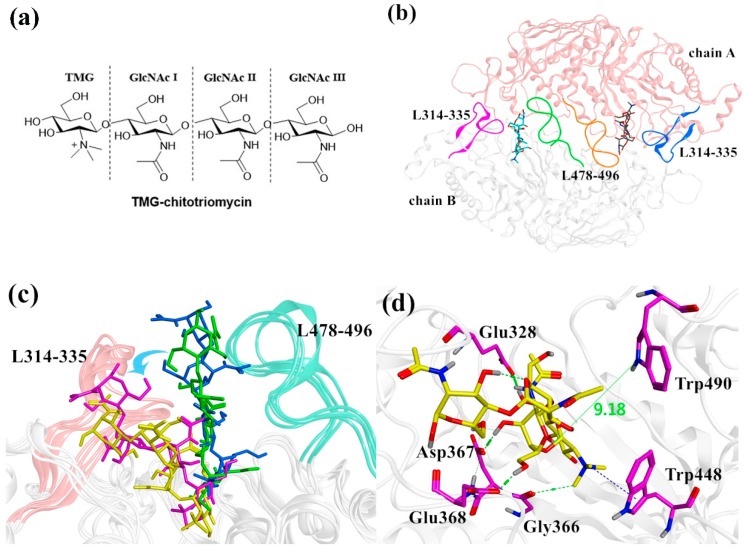
(**a**) The structure of *N,N,N*-trimethyl-d-glucosamine-chitotriomycin (TMG-chitotriomycin). (**b**) The homodimer structure of the TMG-chitotriomycin-O*f*Hex1 complex. The loop L478-496 is stable because of the numerous interactions. (**c**) Structure comparison of conventional molecular dynamics (CMD)1 (yellow), CMD2 (blue), CMD3 (purple), and crystal complex (green). (**d**) A close view of TMG-chitotriomycin bound in O*f*Hex1 from CMD1. Important residues are marked in purple.

**Figure 2 ijms-20-01516-f002:**
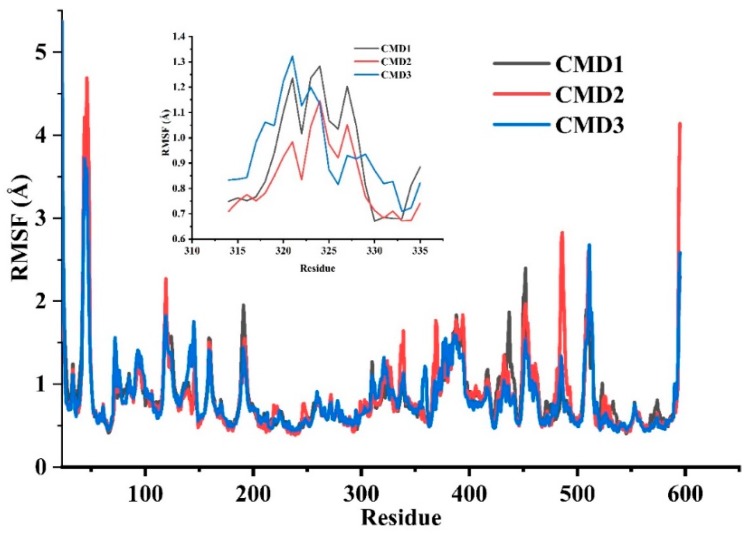
Root mean square fluctuations (RMSF) of an individual residue of O*f*Hex1 for three MD simulation repeats. CMD1 in black, CMD2 in red, CMD3 in blue.

**Figure 3 ijms-20-01516-f003:**
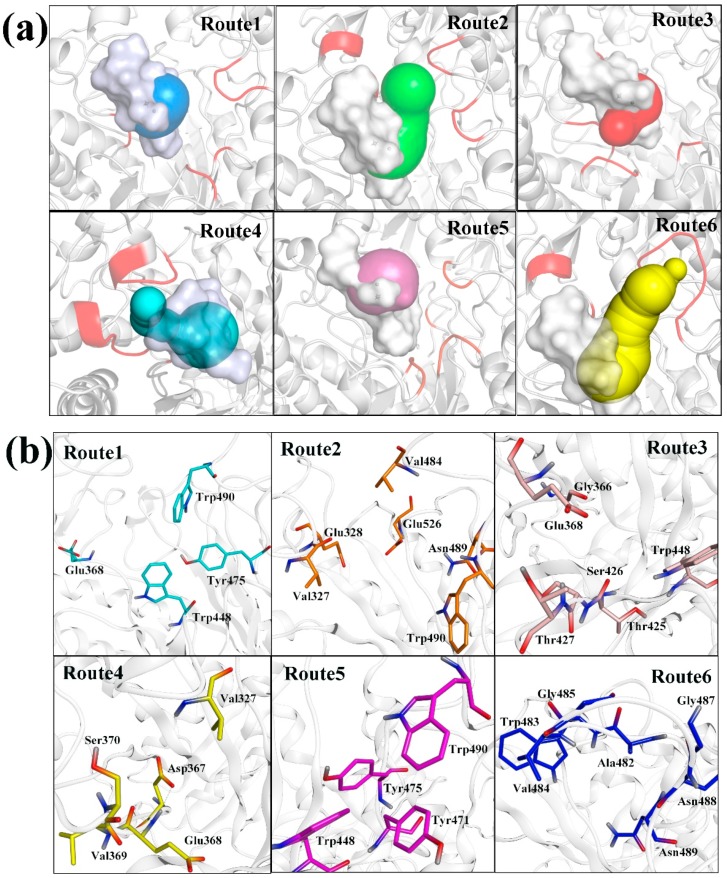
(**a**) Unbinding routes of TMG-chitotriomycin from O*f*Hex1 identified by CAVER from the last snapshot on the 125th nanosecond. (**b**) Residues of tunnels for Route 1, Route 2, Route 3, Route 4, Route 5, and Route 6.

**Figure 4 ijms-20-01516-f004:**
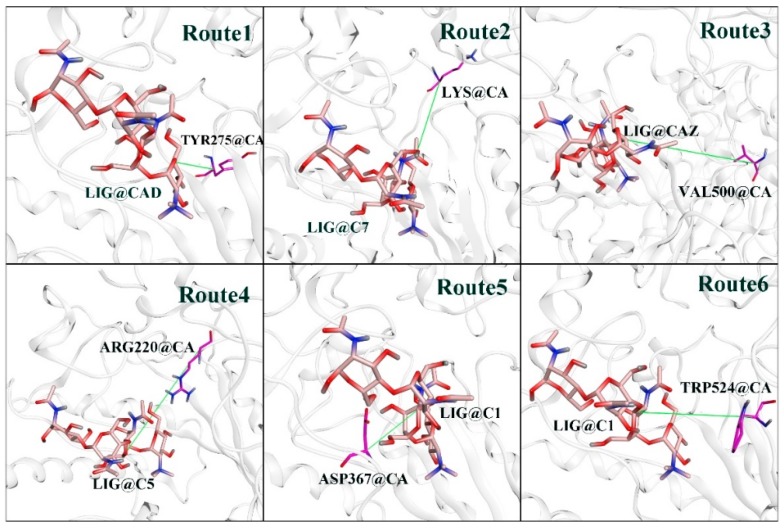
The unbinding direction of all routes in SMD simulations. Route 1: from the Cα atom of Lys273 to the C7 atom of the ligand; Route 2: from the Cα atom of Val500 to the CAZ atom of the ligand; Route 3: from the Cα atom of Val500 to the CAZ atom of the ligand; Route 4: from the Cα atom of Agr220 to the C5 atom of the ligand; Route 5: from the Cα atom of Asp367 to the C1 atom of the ligand; Route 6: from the Cα atom of Trp524 to the C1 atom of the ligand.

**Figure 5 ijms-20-01516-f005:**
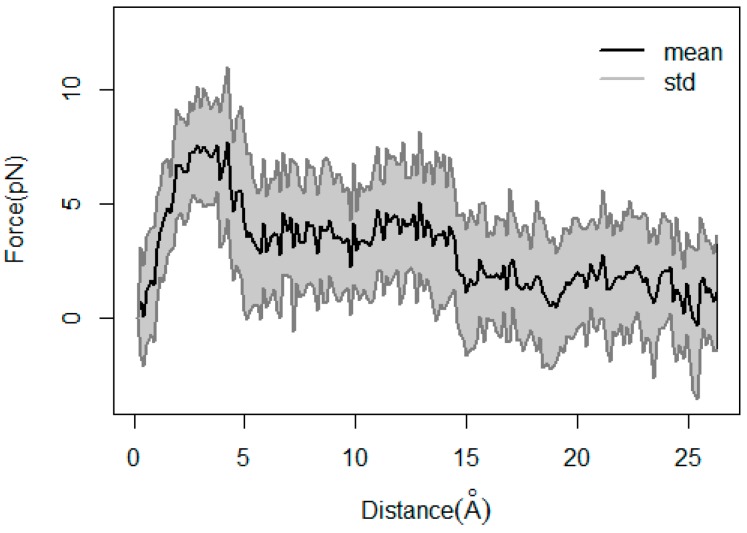
The average force versus the distance of ligand dissociation from the receptor in Route 1.

**Figure 6 ijms-20-01516-f006:**
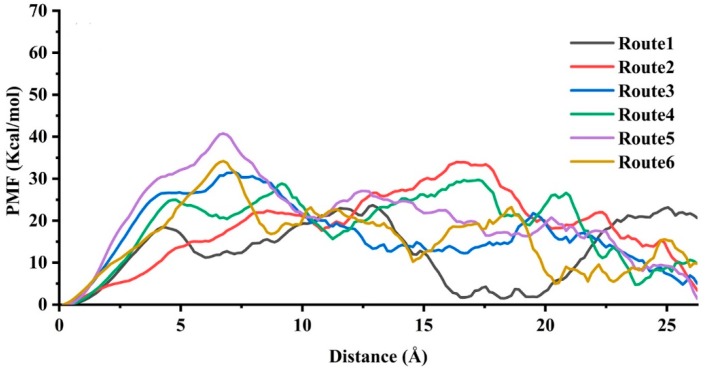
Potential of mean forces (PMFs) of TMG-chitotriomycin dissociation from O*f*Hex1 in different directions versus the distance.

**Figure 7 ijms-20-01516-f007:**
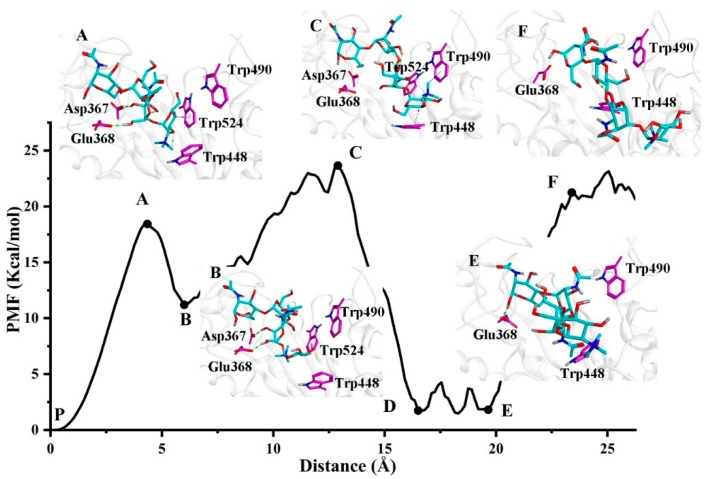
Structural characteristics of TMG-chitotriomycin O*f*Hex1 along the most possible unbinding route (Route 1).

**Figure 8 ijms-20-01516-f008:**
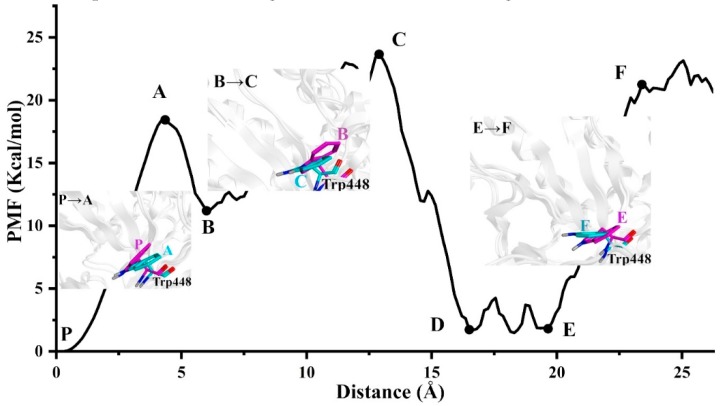
Structural features of Trp448 along the most possible unbinding route.

**Table 1 ijms-20-01516-t001:** Statistical summary of CAVER analysis of TMG-chitotriomycin unbinding from O*f*Hex1.

Dissociation Routes	Average Radius	Average Length	Priority
Route 1	4.23	2.90	0.962
Route 2	3.30	9.14	0.906
Route 3	2.53	9.21	0.876
Route 4	2.34	19.51	0.806
Route 5	2.29	5.90	0.879
Route 6	1.35	15.65	0.730

**Table 2 ijms-20-01516-t002:** The probability of tunnels for Route 1, Route 2, Route 3, Route 4, Route 5, and Route 6.

Route	Residues	Probability (%)
1	Tyr475	11
	Trp490	14
	Glu368	11
	Trp448	14
2	Val484	10
	Asn489	11
	Trp490	11
	Glu526	11
	Val327	11
	Glu328	10
3	Gly366	11
	Glu368	14
	Thr425	14
	Ser426	14
	Thr427	13
	Trp448	14
4	Val327	19
	Asp367	17
	Glu368	20
	Val369	14
	Ser370	13
5	Tyr471	23
	Tyr475	14
	Trp490	27
	Trp448	27
6	Ala482	12
	Gly485	11
	Gly487	10
	Asn488	11
	Trp483	9
	Val484	10
	Asn489	9

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
