# Peer review of "Computational Study for the Unbinding Routes of β-*N*-Acetyl-d-Hexosaminidase Inhibitor: Insight from Steered Molecular Dynamics Simulations"

_ijms, 2019, doi:10.3390/ijms20061516_

Round 1
Reviewer 1 Report
The manuscript (MS) authored by S. Hu, X. Zhao and L. Zhang presents an interesting study on the unbinding routes of a relevant inhibitor of OfHex1 enzyme, a model target for design of insecticides. However, in its present form the MS has drawbacks that must be addressed in a major revision and that are specified below.
The discussion on the structural and binding features of OfHex1 in the Introduction (lines 43-52) should be much clearer if a proper image displaying the structure of the enzyme had been presented.
The Results section starts at discussing structural issues regarding monomers without having previously presented the structural nature of the protein being studied. The reader does not need to know the details of that protein: the authors have to present them. A proper image is again mandatory here.
Subsection 2.1 begins presenting putative dissociation directions of the ligand without any support or rationalization of the assumptions and goes ahead discussing motions observed in the MD trajectories and even “spontaneous unbinding” of the ligand without supporting evidence. In this regard, it is extremely hard to see something of interest in Figure 1.
How is it possible to discuss about differences in “the region of L314-315” (line 101) on the basis of RMSD plots analyzed in the paragraph lines 97-106? Surprisingly, the authors state that changes in that region “might be attributed to the dissociation of TMG-chitotriomycin” (lines 102-103) but apparent proofs of this dissociation are presented in the next paragraph without being announced at its point. If the authors wish to analyze the role of particular residues (or segments) the analysis of RMSF plots is mandatory here. These curves cannot be relegated to supplementary figures.
The last part of section 2.1 is plagued with inconsistencies and a chaotic organization. How was the average structure in Fig. 2a computed from the last 5 ns of the CMD1 simulation? Why was CMD1 selected? What type of protein-ligand interaction energies are represented in Fig. 2b and how were they calculated? (no explanation on this in Material and Methods).
Figure 3 for presenting CAVER results is not very illustrative. CAVER provides an interesting plugin for PyMOL for rendering proper images of tunnel-like cavities found in proteins (see below). The same comment is applicable to the presentation of the routes used in SMD simulations. It is hard to see how those routes are inside the protein by looking at figure 3.
For readers who are not familiar with MD but are interested in the molecular system studied or in OfHex1-like inhibitors, it should be necessary to present a brief introduction of what it the type of information provided by SMD simulations and by PMFs calculated from them. This way, subsections 2.3 and 2.4 should be much more informative.
In subsection 2.5, the presence of several tryptophans near the ligand is very interesting. In the text of this subsection, the authors repeatedly refer to “hydrophobic interactions”, yet in figure 7 it seems that some Trp residues could be forming pi-stacking interactions between their own rings when proper side-chain conformations allow for that (Trp490 and Trp524 or Trp524 and Trp448 in A and B, for example). However, the authors mention a stabilization through …”pi-pi stacking between TMG ring of ligand and the indole ring of Trp448” (lines 205-206). As its name implies, that stacking forms between aromatic rings with pi-electron distributions but the ligand has no aromatic rings (Fig. S1). By the way, figure 7 should benefit from removing main chain atoms in the residues displayed.
The first paragraph in Materials and Methods must be removed.
In subsection 3.2, it is stated that the complex was placed in a 10 Angstrom cubic water box (line 251). This must be a mistake: in a cubic box so small virtually nothing can be placed.
In subsection 3.4, the velocity and force constant settings are given without explanation. The number of times SMD simulations were repeated to calculate PMFs is also given without explanation. In this same subsection, it is stated that …” it was hard to determine atoms in CAVER” (line 274): the authors would probably be willing to take a look at the information found in the web by searching in Google “caver plugin pymol”.
In subsection 3.5, Jarzynski’s equality is correctly presented in terms of Helmholtz free energy. Although Jarzynski’s equality holds for NPT and NVT ensembles, if one is interested in Helmholtz free energy, the initial microstates of the system must be sampled in the NVT ensemble but all the SMD simulations have been performed in the NPT ensemble. This issue must be discussed.
The whole text needs a serious revision for improving the English and above all, for a better organizatio.
Author Response
Point 1: The discussion on the structural and binding features of OfHex1 in the Introduction (lines 43-52) should be much clearer if a proper image displaying the structure of the enzyme had been presented.
Response 1: Thanks for the good advice. We have added the binding pocket of OfHex1 and TMG-chitotriomycin in Figure S1(a) and the 3D structure of TMG-chitotriomycin in Figure S1(b) in Supporting information.
Point 2: The Results section starts at discussing structural issues regarding monomers without having previously presented the structural nature of the protein being studied. The reader does not need to know the details of that protein: the authors have to present them. A proper image is again mandatory here.
Response 2: Thanks for the good suggestion. The first paragraph of section 2.1 was not clear enough and we have rewritten the first paragraph. Please see lines 93-106.
Point 3: Subsection 2.1 begins presenting putative dissociation directions of the ligand without any support or rationalization of the assumptions and goes ahead discussing motions observed in the MD trajectories and even “spontaneous unbinding” of the ligand without supporting evidence. In this regard, it is extremely hard to see something of interest in Figure 1
Response 3: Thanks for the careful question. We have rewritten the second paragraph of section 2.1 and redraw the Figure 1. Please see Figure 1 and Figure S2 and lines 107-123.
Point 4: How is it possible to discuss about differences in “the region of L314-315” (line 101) on the basis of RMSD plots analyzed in the paragraph lines 97-106? Surprisingly, the authors state that changes in that region “might be attributed to the dissociation of TMG-chitotriomycin” (lines 102-103) but apparent proofs of this dissociation are presented in the next paragraph without being announced at its point. If the authors wish to analyze the role of particular residues (or segments) the analysis of RMSF plots is mandatory here. These curves cannot be relegated to supplementary figures.
Response 4: Thanks for the careful question. The discussion of “the region of L314-335” was not based on RMSD, we are sorry to make you misunderstand. We have revised this section. Please see lines 141-161. We have added Figure 2 and Figure S5 as the analysis of RMSF in the manuscript and Figures S3-S5 as the analysis of RMSD in the Supporting information.
Point 5: The last part of section 2.1 is plagued with inconsistencies and a chaotic organization. How was the average structure in Fig. 2a computed from the last 5 ns of the CMD1 simulation? Why was CMD1 selected? What type of protein-ligand interaction energies are represented in Fig. 2b and how were they calculated? (no explanation on this in Material and Methods).
Response 5: Thanks for the careful question. Protein-ligand interaction energies were calculated by MM-PBSA. Considering the energy contribution of the important residues for CMD1 was abrupt in the last part of section 2.1. We have deleted this part and explained why CMD1 was selected. Please see lines 107-161.
Point 6: Figure 3 for presenting CAVER results is not very illustrative. CAVER provides an interesting plugin for PyMOL for rendering proper images of tunnel-like cavities found in proteins (see below). The same comment is applicable to the presentation of the routes used in SMD simulations. It is hard to see how those routes are inside the protein by looking at figure 3.
Response 6: Thanks for the good suggestion. We have downloaded the plugin and redrawn Figure 3.
Point 7: For readers who are not familiar with MD but are interested in the molecular system studied or in OfHex1-like inhibitors, it should be necessary to present a brief introduction of what it the type of information provided by SMD simulations and by PMFs calculated from them. This way, subsections 2.3 and 2.4 should be much more informative.
Response 7: Thanks for the good advice. We have added more information for SMD simulation and PMF calculation in sections 2.3, 2.4, and 3.5. Please see lines 230-293, and 423-435.
Point 8: In subsection 2.5, the presence of several tryptophans near the ligand is very interesting. In the text of this subsection, the authors repeatedly refer to “hydrophobic interactions”, yet in figure 7 it seems that some Trp residues could be forming pi-stacking interactions between their own rings when proper side-chain conformations allow for that (Trp490 and Trp524 or Trp524 and Trp448 in A and B, for example). However, the authors mention a stabilization through …”pi-pi stacking between TMG ring of ligand and the indole ring of Trp448” (lines 205-206). As its name implies, that stacking forms between aromatic rings with pi-electron distributions but the ligand has no aromatic rings (Fig. S1). By the way, figure 7 should benefit from removing main chain atoms in the residues displayed.
Response 8: Thanks for the careful question. We are sorry that we made a mistake on the interactions between TMG ring of the ligand and the indole ring of Trp524, Trp448 and Trp490. The interactions should be H-π interactions. We have revised them. Please see lines 315, 319, and 323. We have deleted main chain atoms to draw Figure 7 again. Please see Figure 8.
Point 9: The first paragraph in Materials and Methods must be removed.
Response 9: Thanks for the careful advice. We have removed the first paragraph in Materials and Methods. Please see lines 353-358.
Point 10: In subsection 3.2, it is stated that the complex was placed in a 10 Angstrom cubic water box (line 251). This must be a mistake: in a cubic box so small virtually nothing can be placed.
Response 10: Thanks for the careful questions. The previous expression was wrong and we have revised it. Please see lines 375-390.
Point 11: In subsection 3.4, the velocity and force constant settings are given without explanation. The number of times SMD simulations were repeated to calculate PMFs is also given without explanation. In this same subsection, it is stated that …” it was hard to determine atoms in CAVER” (line 274): the authors would probably be willing to take a look at the information found in the web by searching in Google “caver plugin pymol”.
Response 11: Thanks for the good advice. We have added the related explanation in section 2.3. Please see lines 255-261. We have downloaded the plugin and redrawn the picture with PyMOL. Please see Figure 3.
Point 12: In subsection 3.5, Jarzynski’s equality is correctly presented in terms of Helmholtz free energy. Although Jarzynski’s equality holds for NPT and NVT ensembles, if one is interested in Helmholtz free energy, the initial microstates of the system must be sampled in the NVT ensemble but all the SMD simulations have been performed in the NPT ensemble. This issue must be discussed.
Response 12: Thanks for the important question. We have added new information for MD simulations and SMD simulations in section 3.2 and section 3.4. Please see lines 375-390 and lines 402-403.
Reviewer 2 Report
At first, I thank authors for carrying out this interesting research. However, I will accept this manuscript for publishing if the authors improve the article and clarify comments below.
1-In lines 59 and 60, the author states that the dissociation rate can be simulated by SMD. However, SMD does not result in rate. SMD generates potential mean forces which can be used to indirectly perceive ligand binding strength.
1-Please site the following paper in the lines 61 and 62: Insulin mimetic peptide S371 folds into a helical structure, https://onlinelibrary.wiley.com/doi/full/10.1002/jcc.24746, refer to Figure S3 in supplemental material in this paper to see how insulin (as a ligand) is dissociated from its pocked by SMD simulation. You may also cite this PhD dissertation: “Simulation Studies of Signaling and Regulatory Proteins”.
2- In Figure 1, the large part of the figure does not contain informative image. It is highly recommended to focus on the part of interest. In addition, please make a nice cartoon of the inhibitor (TMG-chitotriomycin) and show hydrogen bonds and important features using figure. Be specific about the residues and highlight the residues that have crucial interactions with the ligand.
3- Please provide information about the ligand-receptor distance fluctuations during the three simulations. You may plot distance of COMs between the ligand and the receptor versus time.
4-In lines 133 to 143, the author used CAVER tools to probe different routes for dissociation of ligand from the pocket. It is necessary to point out the algorithm and the scientific approach by which these routes have been selected and clarify what the priority values are. No software and tools should be used as a black-box, but the method and basis should be pointed out and cited.
2-Figure 6 shows PMFs for different routes. PMF is a state function means it does not have dependency to the path of the process, it is merely dependent of the start and end points. How do you illustrate this conflict between PMF as a state function and different PMFs for routes? Please explain and clarify this in the manuscript. You may also cite this PhD dissertation for clarification “Simulation Studies of Signaling and Regulatory Proteins”.
5-The PMFs and forces are shown versus time. It is highly recommended to show PMFs and forces with respect to distance of ligand from the receptor.
Author Response
Point 1: In lines 59 and 60, the author states that the dissociation rate can be simulated by SMD. However, SMD does not result in rate. SMD generates potential mean forces which can be used to indirectly perceive ligand binding strength.
Response 1: Thanks for the important question. We are sorry that our previous expression “The dissociation rate can be simulated by the steered molecular dynamics (SMD) with low computational cost and reproducibility.” was wrong. We have rewritten this section. Please see lines 63-74.
Point 2: Please site the following paper in the lines 61 and 62: Insulin mimetic peptide S371 folds into a helical structure, https://onlinelibrary.wiley.com/doi/full/10.1002/jcc.24746, refer to Figure S3 in supplemental material in this paper to see how insulin (as a ligand) is dissociated from its pocked by SMD simulation. You may also cite this PhD dissertation: “Simulation Studies of Signaling and Regulatory Proteins”.
Response 2: Thanks for the valuable advice. The paper and PhD dissertation are very useful for our study and we have cited the paper and PhD dissertation in the manuscript. Please see line 66.
Point 3: In Figure 1, the large part of the figure does not contain informative image. It is highly recommended to focus on the part of interest. In addition, please make a nice cartoon of the inhibitor (TMG-chitotriomycin) and show hydrogen bonds and important features using figure. Be specific about the residues and highlight the residues that have crucial interactions with the ligand.
Response 3: Thanks for the good advice. We have redrawn Figure 1.
Point 4: Please provide information about the ligand-receptor distance fluctuations during the three simulations. You may plot distance of COMs between the ligand and the receptor versus time.
Response 4: Thanks for the good advice. We have added this Figure and discussed it. Please see lines 118-121 and Figure S2.
Point 5: In lines 133 to 143, the author used CAVER tools to probe different routes for dissociation of ligand from the pocket. It is necessary to point out the algorithm and the scientific approach by which these routes have been selected and clarify what the priority values are. No software and tools should be used as a black-box, but the method and basis should be pointed out and cited.
Response 5: Thanks for the valuable advice. We have revised it in sections 2.2 and 3.4. Please see lines 189-198 and 402-403.
Point 6: Figure 6 shows PMFs for different routes. PMF is a state function means it does not have dependency to the path of the process, it is merely dependent of the start and end points. How do you illustrate this conflict between PMF as a state function and different PMFs for routes? Please explain and clarify this in the manuscript. You may also cite this PhD dissertation for clarification “Simulation Studies of Signaling and Regulatory Proteins”.
Response 6: Thanks for the important question. We have revised it in section 3.5 and redrawn Figure 7 using PMF versus the distance. Please see Figure 7 and lines 423-435.
Point 7: The PMFs and forces are shown versus time. It is highly recommended to show PMFs and forces with respect to distance of ligand from the receptor.
Response 7: Thanks for the good advice. We have redrawn Figure 5 and Figure 6 using the distance instead of time.
Reviewer 3 Report
The objective of this work is interesting and well-intended. It pursuits to generate computational data that will provide understanding about the unbinding routes of ligands from the OfHex1 protein. However, it does not succeed at presenting clear results that (1) fully support their conclusions and (2) give insightful answers to many questions raised all over the text. A number of serious problems emerge throughout:
- The whole paper needs to be rethought in order to avoid (1) fundamental errors, (2) results overinterpretation and (3) overstatements: considering of 150ns CMD as "long-time MD" or citing MD as a minimization technique, or repeating MD "by setting up different velocities", etc.
- Conformational and positional changes require further structural descriptions than those provided by terms like "straight out inside", "downward" or "away". It is mandatory to provide better and more detailed data.
- New quantitative analyses must be performed over the existing CMD trajectories. For instance, RMSF/RMSD analyses performed on the simulations are (1) insufficient and (2) not correctly described. Detailed reasoning on the MD protocols and Jarzynski integration should be also explained. PMFs calculated from irreversible pulling are more insightful if error bars (SE or SD) are provided over the average work (especially when authors compare different unbinding routes).
- Figures' quality and the lack of details in their annotations are hardly acceptable in their current state. Unfortunately, it seems that submission has been rushed beyond desirable. For example, the first paragraph in the Materials and Methods section does not correspond to any kind of research. Rather it seems to be editorial instructions for authors.
Overall, the Authors have aimed their project in the right direction. However the work presented here has numerous issues that need to be fixed/reconsidered before getting their manuscript published. All in all, this Referree encourages the Authors to resubmit a mature version after a complete and profound revision.
Author Response
Point 1: The whole paper needs to be rethought in order to avoid (1) fundamental errors, (2) results overinterpretation and (3) overstatements: considering of 150ns CMD as "long-time MD" or citing MD as a minimization technique, or repeating MD "by setting up different velocities", etc.
Response 1: Thanks for the important advice. We have deleted overexpression words and added the important and detail discussion for the whole manuscript. Please see the track changes.
Point 2: Conformational and positional changes require further structural descriptions than those provided by terms like "straight out inside", "downward" or "away". It is mandatory to provide better and more detailed data.
Response 2: Thanks for the good advice. We have revised it with more detail data and redrawn Figure1. Please see lines 93-123 and Figure 1.
Point 3: New quantitative analyses must be performed over the existing CMD trajectories. For instance, RMSF/RMSD analyses performed on the simulations are (1) insufficient and (2) not correctly described. Detailed reasoning on the MD protocols and Jarzynski integration should be also explained. PMFs calculated from irreversible pulling are more insightful if error bars (SE or SD) are provided over the average work (especially when authors compare different unbinding routes).
Response 3: Thanks for the valuable advice. 1) We have revised RMSF/RMSD analyses. Please see lines 141-161. We have added Figure 2 and Figure S5 as the analysis of RMSF in the manuscript and Figures S3-S5 as the analysis of RMSD in the Supporting information. 2) We have added the more detail information for MD simulation and PMFs calculation. Please see lines 375-390 and lines 423-435. 3) We have revised the SMD discussion and provided the error bars of the average force and work versus the distance. Please see lines 255-279, Figure 5 and Figure 6.
Point 4: Figures' quality and the lack of details in their annotations are hardly acceptable in their current state. Unfortunately, it seems that submission has been rushed beyond desirable. For example, the first paragraph in the Materials and Methods section does not correspond to any kind of research. Rather it seems to be editorial instructions for authors.
Response 4: Thanks for the good advice. We have redrawn all figures except for Figure 8 and revised the corresponding expression. Please see Figures 1-7, Figure 9 in the manuscript and Figure S1, Figure S2 and Figure S5 in the Supporting information. We have also removed the first paragraph of the Materials and Methods.
Reviewer 4 Report
Comments and Suggestions for Authors
The manuscript entitled "Computational study for the unbinding routes of 3β‐N‐Acetyl‐D‐ hexosaminidase inhibitor: insight from 4 steered molecular dynamics simulations" by Song Hu and etal, studied the dissociation behaviour of inhibitor TMG‐chitotriomycin of OfHex1 by CMD and SMD simulations. The results of CMD revealed that TMG‐chitotriomycin exhibited a dissociation trend away from active pocket. Six different routes for unbinding of TMG‐chitotriomycin from OfHex1 were explored.
In my opinion, the manuscript is good written and valuable for readers of IJMS after revising the English languish and the typographical errors.
Author Response
Point 1: In my opinion, the manuscript is good written and valuable for readers of IJMS after revising the English languish and the typographical errors.
Response 1: Thank you for the good advice. We have invited English native editor to edit our manuscript. Please see the certificate letter.
Round 2
Reviewer 1 Report
In this revised version, the authors have addressed satisfactorily most of the concerns raised in my previous review of this manuscript (MS). However, there still remains a large number of mistakes and errors in the formal presentation of the MS as well as in the whole text that needs a serious revision for improving the English.
For example, the sentences in lines 67-68 and 68-69 are repeated. Some statements in the new text added are unintelligible (e.g. those in lines 78-82 or those in lines 168-171).
Figures 3A and S6 are apparently the same. The text in lines 134-135 states that these two figures are compared concluding that the dissociation directions are the same but the meaning of those two figures is not explained.
If the temperature considered is 298 K, the value of kB T in line 315 is wrong.
Summarizing, the MS presents an interesting work but in its current form, it still needs a revision. In particular, the authors are encouraged to drastically correct the English.
Author Response
Dear Reviewer,
Thank you for the valuable and constructive comments from you. According to the recommendations, all questions and suggestions have been answered and considered, and the point-to-point responses to the reviewers’ comments are detailed below in the revision notes. We have invited the native English scientist to edit our revised manuscript shown as the Certificate letter.
Point 1: The sentences in lines 67-68 and 68-69 are repeated. Some statements in the new text added are unintelligible (e.g. those in lines 78-82 or those in lines 168-171).
Response 1: Thanks for the careful advice. We have deleted the repeated sentences in lines 68-69. About the source of homodimer structure, we have revised our statements. Please see lines 88-93. About the selection of velocity and force constant, we have revised our expression. Please see lines 180-185.
Point 2: Figures 3A and S6 are apparently the same. The text in lines 134-135 states that these two figures are compared concluding that the dissociation directions are the same but the meaning of those two figures is not explained.
Response 2: Thanks for the good question. Figure 3a shows the top six routes determined by CAVER from the last snapshot on the 125th nanosecond. Figure S6 shows the top six routes clustered by CAVER from 100 snapshots of the last 5ns. We have revised the captions of Figures 3a and S6. We have also revised the statements in the manuscript. Please see lines 143-145.
Point 3: If the temperature considered is 298 K, the value of kB T in line 315 is wrong.
Response 3: Thanks for the careful question. We are sorry that we made a mistake on the temperature. In fact, we made the simulation at 310K not 298K. We have revised it in the manuscript. Please see lines 287 and 303.
Reviewer 3 Report
The whole paper has been profusely modified. The Authors have corrected most of the flaws in first manuscript. However, a number of small issues are pending for correction.
92-97 & S2. What atom(s) are used to compute "(...) the distance between the ligand and Trp490 during (...)"? Atom mask(s) used for this calculation should be provided.
175-176. average work +/-SD (or SE) should be provided in the text. Figure 5 should be moved to supplementary material.
317-322. Authors did not estimate residence times for this system. This paragraph cannot be included in 'Conclusions' section because it could suggest wrong interpretations to readers unexpert in the field. Rather, it would be more appropriate to reduce its extension and include it as part of the Introduction.
Author Response
Point 1: 92-97 & S2. What atom(s) are used to compute "(...) the distance between the ligand and Trp490 during (...)"? Atom mask(s) used for this calculation should be provided.
Response 1: Thanks for the good advice. We used the O4 atom of ligand and NE1 atom of Trp490 to compute the distance. We have revised it. Please see lines 98-99 in the manuscript and Figure S2 in Supporting information.
Point 2: 175-176. average work +/-SD (or SE) should be provided in the text. Figure 5 should be moved to supplementary material.
Response 2: Thanks for the good suggestion. We have added the average work +/-SD in the manuscript. Please see lines 189-190. Figure 5 have been moved to Supporting information as Figure S7.
Point 3: 317-322. Authors did not estimate residence times for this system. This paragraph cannot be included in 'Conclusions' section because it could suggest wrong interpretations to readers unexpert in the field. Rather, it would be more appropriate to reduce its extension and include it as part of the Introduction.
Response 3: Thanks for the good advice. We have deleted the statements of residence times in Conclusions and added the similar and concise expression in Introduction. Please see lines 56-57.
Round 3
Reviewer 1 Report
The authors have satisfactorily addressed the minor concerns raised in my latter review